# Investigating the Molecular Mechanisms of Resveratrol in Treating Cardiometabolic Multimorbidity: A Network Pharmacology and Bioinformatics Approach with Molecular Docking Validation

**DOI:** 10.3390/nu16152488

**Published:** 2024-07-31

**Authors:** Wei Gong, Peng Sun, Xiujing Li, Xi Wang, Xinyu Zhang, Huimin Cui, Jianjun Yang

**Affiliations:** 1Public Health School, Ningxia Medical University, Yinchuan 750004, China; gongwei@nxmu.edu.cn (W.G.);; 2Key Laboratory of Environmental Factors and Chronic Disease Control, Yinchuan 750004, China; 3School of Medical Information and Engineering, Ningxia Medical University, Yinchuan 750004, China; 4Science and Technology Center, Ningxia Medical University, Yinchuan 750001, China; 5Ningxia Hui Autonomous Region Institute of Medical Sciences, Ningxia Medical University, Yinchuan 750004, China; 6School of Pharmacy, Ningxia Medical University, Yinchuan 750004, China

**Keywords:** resveratrol, cardiometabolic multimorbidity, network pharmacology, molecular docking, molecular dynamics simulation

## Abstract

Background: Resveratrol is a potent phytochemical known for its potential in treating cardiometabolic multimorbidity. However, its underlying mechanisms remain unclear. Our study systematically investigates the effects of resveratrol on cardiometabolic multimorbidity and elucidates its mechanisms using network pharmacology and molecular docking techniques. Methods: We screened cardiometabolic multimorbidity-related targets using the OMIM, GeneCards, and DisGeNET databases, and utilized the DSigDB drug characterization database to predict resveratrol’s effects on cardiometabolic multimorbidity. Target identification for resveratrol was conducted using the TCMSP, SymMap, DrugBank, Swiss Target Prediction, CTD, and UniProt databases. SwissADME and ADMETlab 2.0 simulations were used to predict drug similarity and toxicity profiles of resveratrol. Protein–protein interaction (PPI) networks were constructed using Cytoscape 3.9.1 software. Gene Ontology (GO) and Kyoto Encyclopedia of Genes and Genomes (KEGG) functional enrichment analyses were performed via the DAVID online platform, and target-pathway networks were established. Molecular docking validated interactions between core targets and resveratrol, followed by molecular dynamics simulations on the optimal core proteins identified through docking. Differential analysis using the GEO dataset validated resveratrol as a core target in cardiometabolic multimorbidity. Results: A total of 585 cardiometabolic multimorbidity target genes were identified, and the predicted results indicated that the phytochemical resveratrol could be a major therapeutic agent for cardiometabolic multimorbidity. SwissADME simulations showed that resveratrol has potential drug-like activity with minimal toxicity. Additionally, 6703 targets of resveratrol were screened. GO and KEGG analyses revealed that the main biological processes involved included positive regulation of cell proliferation, positive regulation of gene expression, and response to estradiol. Significant pathways related to MAPK and PI3K-Akt signaling pathways were also identified. Molecular docking and molecular dynamics simulations demonstrated strong interactions between resveratrol and core targets such as MAPK and EGFR. Conclusions: This study predicts potential targets and pathways of resveratrol in treating cardiometabolic multimorbidity, offering a new research direction for understanding its molecular mechanisms. Additionally, it establishes a theoretical foundation for the clinical application of resveratrol.

## 1. Introduction

Cardiometabolic multimorbidity (CMM) is one of the most common patterns of multimorbidity among the elderly population, encompassing the co-occurrence of two or more cardiovascular and metabolic conditions such as ischemic stroke (IS), coronary heart disease (CHD), and diabetes mellitus (DM) [1]. Currently, the prevalence of chronic diseases among the elderly population in China is reported to be 42.4%, with CMM affecting approximately 11.61% [2]. CMM significantly impacts life expectancy, reducing it by 12 years in 60-year-olds with two diseases and by 15 years in those with three diseases compared to individuals without any cardiometabolic conditions [2]. Given the global demographic shift towards an aging population, CMM has become a significant global health concern [3]. This underscores the importance of intervention research and metabolism exploration in CMM in the elderly population.

Recent studies have demonstrated the potential benefits of dietary interventions and nutritional supplements in reducing cardiovascular morbidity and mortality. Additionally, there is growing interest in the therapeutic use of natural products and active ingredients found in Chinese herbs, such as ginseng, ginkgo biloba, and Ganoderma lucidum, for the treatment of CMM [4,5]. However, multi-target natural products with significant pharmacological activity offer a greater advantage [6].

Resveratrol (3,5,4′-trihydroxytrans-stilbene) is a member of the stilbene group within the polyphenol family, naturally occurring in various plants such as nuts, berries, and grapes. The compound exhibits antioxidant properties [7], anti-inflammatory effects [8], enhancement of mitochondrial function [9], neuroprotective capabilities, and potentially possesses anti-aging and immune-modulating effects [10]. Resveratrol has garnered considerable attention for its efficacy in treating cardiovascular diseases. A randomized double-blind clinical trial (RCT) has demonstrated that resveratrol offers benefits for heart failure [11] and exhibits a protective effect against atherosclerosis among individuals at low cardiovascular risk [12]. Moreover, resveratrol is widely utilized in diabetes treatment, primarily for its roles in reducing blood sugar levels, enhancing insulin sensitivity, and preserving pancreatic beta cells [13]. Clinical trials have highlighted that resveratrol supplementation can effectively improve blood glucose control and insulin sensitivity, making it a potential adjunct in diabetes management [7]. Nonetheless, the precise mechanism underlying the therapeutic effects of resveratrol on CMM remains elusive.

Network pharmacology is an innovative approach in pharmacological research rooted in systems biology, transcending the constraints of traditional single-target drug studies. It elucidates that drugs typically act on multiple targets and that these targets often intersect with various diseases. This methodology has been instrumental in predicting protein targets of active plant ingredients and their associated pathological pathways [8]. Currently, there are limited studies on resveratrol for the treatment of CMM using network pharmacology.

The main objectives of this study are as follows:Utilize the OMIM, GeneCards, and DisGeNET databases to identify targets associated with CMM.Incorporate the DSigDB drug characterization database into enrichment analyses to predict drug candidates for addressing CMM.Identify active compounds and their corresponding targets in resveratrol using the TCMSP, SymMap, DrugBank, Swiss Target Prediction, CTD, and UniProt databases.Validate drug similarities and toxic properties using the SwissADME and ADMETlab 2.0 web tools.Explore interactions among common targets of resveratrol and CMM using PPI databases, and construct a PPI network using Cytoscape 3.9.1 software.Conduct enrichment analysis to assess the distribution of resveratrol’s targets in diseases, cellular components (CCs), biological processes (BPs), molecular functions (MFs), and signaling pathways.Develop a target-pathway topological network model to elucidate the mechanisms of resveratrol against CMM.Evaluate interactions between resveratrol and core targets through molecular docking and molecular dynamics simulations.Perform differential analysis of GEO datasets to identify potential core targets of resveratrol in addressing CMM.

In this study, we explored the mechanism of resveratrol in modulating CMM through network pharmacology. We employed molecular docking and molecular dynamics simulations to assess the interaction properties between resveratrol and its key targets, validating these insights through GEO datasets. These findings offer novel insights into the therapeutic potential of phytochemicals in managing CMM. They also provide theoretical guidance and a scientific foundation for future experimental investigations and clinical applications. The flow chart of this study is shown in Figure 1.

## 2. Materials and Methods

### 2.1. Identification of CMM Target Genes

CMM encompasses conditions such as IS, CHD, and DM. To identify disease-specific targets, we used keywords like “ischemic stroke”, “coronary heart disease”, and “diabetes mellitus” to query the OMIM (https://www.omim.org/, accessed on 6 March 2024), GeneCards (https://www.genecards.org/, accessed on 6 March 2024), and DisGeNET (https://www.disgenet.org/, accessed on 6 March 2024) databases.

### 2.2. Drug Prediction

The DSigDB database within the Enrichr platform (https://maayanlab.cloud/Enrichr/enrich#, accessed on 6 March 2024) was used to identify drug molecules linked to these multimorbidity genes, applying a filtering criterion of *p* < 0.05. From these analyses, the top 10 drugs were selected as potential candidates for treating CMM.

### 2.3. Drug Similarity and Toxicity Analysis

Drug similarity and toxicity properties were assessed using SwissADME (http://www.swissadme.ch/, accessed on 7 March 2024) and ADMETlab 2.0 (https://admetmesh.scbdd.com/, accessed on 7 March 2024). These analyses followed critical criteria, including Lipinski’s rule parameters: molecular weight (<500 g/mol), hydrogen bond acceptors (<10), hydrogen bond donors (≤5), Moriguchi octanol–water partition coefficient (≤4.15), and topological polar surface area (<140). Additionally, resveratrol’s toxicity was evaluated using ADMETlab 2.0, focusing on five key parameters: hERG blockers, human hepatotoxicity, eye corrosion, respiratory toxicity, and rat oral acute toxicity.

### 2.4. Identification of Resveratrol Target Genes

Target genes for resveratrol were obtained using “resveratrol” as the keyword in the TCMSP (https://old.tcmsp-e.com/tcmsp.php, accessed on 7 March 2024), SymMap (www.symmap.org/, accessed on 7 March 2024), DrugBank (https://go.drugbank.com/, accessed on 7 March 2024), Swiss Target Prediction (https://www.swisstargetprediction.ch/, accessed on 7 March 2024), and CTD (https://ctdbase.org/, accessed on 7 March 2024) databases. The identified targets were then cross-referenced with the UniProt database for “Homo sapiens”.

### 2.5. Protein–Protein Interaction (PPI) Network Analysis

Target genes associated with IS, CHD, DM, and their combinations (IS + CHD, IS + DM, CHD + DM, and IS + CHD + DM) were identified through respective intersections. Redundant genes were removed, and the resulting intersecting genes were combined with the compiled list of resveratrol target genes. Venny2.1 (https://bioinfogp.cnb.csic.es/tools/venny/, accessed on 10 March 2024) was used to visualize these intersections. The identified target genes were further analyzed using the String online platform (https://cn.string-db.org/, accessed on 10 March 2024), specifying “Homo sapiens” as the species and setting a confidence score threshold of 0.7 for PPI network construction. Cytoscape 3.9.1 software, along with the Centiscape2.2 and CytoNCA 2.1.6 plugins, facilitated topology analysis and the creation of PPI networks based on degree values for the common targets.

### 2.6. Gene Ontology and Pathway Analysis

The intersection targets of resveratrol and CMM were analyzed using the DAVID online platform (https://david.ncifcrf.gov/, accessed on 10 March 2024) for Gene Ontology (GO) and Kyoto Encyclopedia of Genes and Genomes (KEGG) pathway enrichment studies. This included investigation into molecular functions (MFs), cellular components (CCs), biological processes (BPs), and KEGG pathways. The top 10 GO entries and top 20 KEGG pathways were selected based on statistical significance (*p* < 0.05) and visualized using the bioinformatics online platforms (http://www.bioinformatics.com.cn/, accessed on 10 March 2024). In the KEGG database, the intersecting genes were subjected to KEGG Mapper analysis, focusing on “Homo sapiens”. Pathway analyses encompassed metabolism, environmental information processing, cellular processes, organismal systems, and human diseases.

### 2.7. MCODE Clustering Analysis

Metascape (https://metascape.org/, accessed on 10 March 2024) was used for clustering similar proteins and building functional modules using molecular complex detection (MCODE). MCODE analysis identified protein relationship complexes and clustered them into gene clusters where each target has the same or similar gene function. Additionally, transcription factors that regulate other targets were predicted, providing information on target interactions. MCODE clustering analysis was conducted for common targets based on a minimum overlap of 3, *p*-value cutoff of 0.05, and minimum enrichment of 1.5.

### 2.8. Network Construction of Target-Pathway Interaction

The top 20 pathways resulting from the KEGG enrichment analysis of resveratrol and CMM were visualized using Cytoscape 3.9.1 software to create a compound–target–pathway network diagram. Network topology parameters were examined, with targets ranked by degree value. The top 20 targets from this analysis were intersected with those identified in the PPI network to pinpoint common targets, thereby identifying the final core action targets.

### 2.9. Molecular Docking

Molecular docking was conducted to predict potential therapeutic effects by assessing the binding energy between drug components and key protein targets. Molecular docking simulations were conducted using Schrödinger software 13.5 (February 2023 version). The active ingredient of resveratrol was retrieved from the TCMSP database in mol2 format. Six core macromolecular proteins were identified from the PDB database (https://www.rcsb.org/, accessed on 15 March 2024), focusing on “Homo sapiens” and employing the minimum method for selecting “Unique Ligands”. Protein structures underwent optimization using Schrödinger’s Protein Preparation Wizard, which included hydrogen bond assignment and restrained minimization. Subsequently, protein structures were analyzed and receptor grids were generated based on native ligand interactions. Ligand docking was performed using Glide v.6.5 software in Standard Precision (SP Docking) mode. The results provided insights into compound–protein interactions, and structural analysis was conducted using Maestro 2023.

### 2.10. Molecular Dynamics

Molecular dynamics (MD) simulations were performed using the Desmond v53011 Dynamics module within Schrödinger software. The protein–ligand complex was prepared using the Protein Prepare module with default parameters, which involved correcting binding information, adding hydrogen atoms, and removing water molecules. The prepared protein was solvated using the TIP3P water model and OPLS3 force field. Sodium ions (Na^+^) were added to simulate physiological saline conditions. MD simulations were performed using default parameters. The Simulation Interactions Diagram function was utilized to analyze output files from the MD module. The root mean square deviation (RMSD) metric measured the average change in atom displacement relative to a reference frame across all structures in the trajectory, providing insights into the stability and dynamics of the protein–ligand complex.

### 2.11. Acquisition and Target Validation from the GEO Database

Datasets for IS (GSE140275), CHD (GSE12288), and DM (GSE29221) from the GEO database (https://www.ncbi.nlm.nih.gov/geo/, accessed on 20 March 2024) underwent differential analysis using the “limma” package in R version 4.1.0 (R Foundation for Statistical Computing, Vienna, Austria). Differential expression was assessed with thresholds set at |logFC| > 1 and *p* < 0.05. Volcano plots and heat maps were generated using the “ggplot2” and “pheatmap” R packages to visualize and analyze the differential gene expression profiles.

## 3. Results

### 3.1. Acquisition of CMM Target Genes

IS, CHD, and DM data were sourced from the OMIM, GeneCards, and DisGeNet databases, respectively. After removing duplicates, we identified 2205 unique targets for IS, 4835 for CHD, and a total of 8331 for DM (Appendix A). These disease targets were then mapped to the UniProt database, yielding 336, 659, and 1069 target genes under the condition of “Homo sapiens” (Appendix A). Overlapping analyses revealed 282 genes common to IS and CHD, 292 genes common to IS and DM, 537 genes common to CHD and DM, and 263 genes common across all three conditions (Appendix A). Finally, the intersection of these overlapping categories was compiled and duplicates were removed, resulting in a final set of 585 CMM genes (Appendix A).

### 3.2. Drug Prediction

We utilized the genes associated with CMM to analyze drugs targeting them using the DSigDB database. Our findings emphasize resveratrol, a phytochemical, as a promising candidate for therapeutic intervention in CMM, as elaborated in Table 1.

### 3.3. Drug Similarity and Toxicity Analysis

The molecular structure of resveratrol (C_14_H_12_O_3_) studied in this paper is presented in Table 2, which is the trans-structure. The SwissADME platform conducted drug similarity analysis of resveratrol, confirming its compliance with Lipinski’s rule and suggesting that it meets pharmacokinetic criteria suitable for new drug evaluation. This underscores its potential as a key ingredient in the development of therapeutic agents (Table 2).

Furthermore, evaluation of resveratrol using the ADMETlab platform assessed parameters including hERG blockers, human hepatotoxicity, ocular corrosion, respiratory toxicity, and oral acute toxicity in rats. The results indicate low toxicity levels, positioning resveratrol as a promising candidate for the treatment of CMM (Table 3).

### 3.4. Acquisition of Resveratrol Targets

Following the component-mining process described above, a total of 6703 resveratrol target genes were identified across the TCMSP, SymMap, DrugBank, Swiss Target Prediction, and CTD databases, focusing on ‘Homo sapiens’ as the specified condition (Appendix A).

### 3.5. PPI Network Analysis

The Venny online platform was utilized to analyze 6703 resveratrol target genes and 585 CMM target genes, resulting in the identification of 324 common genes (Figure 2A, Appendix A).

These shared genes were input into String and used to construct a protein–protein interaction (PPI) network diagram, which comprised 264 nodes and 989 edges. Topological analysis of the PPI network focused on metrics such as node degree centrality, median centrality, and neighboring centrality. Initially, nodes with a degree centrality (DC) ≥ 9 were filtered, resulting in 86 nodes and 534 edges. Further filtering with a DC ≥ 18 retained 20 nodes and 82 edges (Figure 2B, Appendix A). In this context, nodes represent proteins and edges signify their interactions. Nodes with higher degree values indicate greater interactions and importance within the network.

The PPI network diagram highlighted 20 key node proteins, including INS, EGFR, ALB, TNF, STAT3, LEP, IFNG, APOE, and others, which play pivotal roles in the treatment of CMM. The degree values of these nodes are illustrated in Figure 2C, underscoring their significance in network interactions and potential therapeutic relevance.

### 3.6. GO Function and KEGG Pathway Enrichment Analysis

The 324 intersecting genes identified between resveratrol and CMM were analyzed using the DAVID online platform for Gene Ontology (GO) and Kyoto Encyclopedia of Genes and Genomes (KEGG) pathway enrichment. GO analysis resulted in 795 entries, with 572 involved in biological processes (BPs), 92 in cellular components (CCs), and 131 in molecular functions (MFs). BP terms included positive regulation of cell proliferation, gene expression, response to estradiol, transcriptional regulation of RNA polymerase II promoter, and protein phosphorylation. CC terms encompassed extracellular matrix, cell surface, myofibrillar membrane, and platelet α-granule lumen, while MF terms included protein binding, growth factor activity, RNA polymerase II transcription factor activity, and ligand-activated sequence-specific DNA binding. The top 10 significant entries (*p* < 0.05) were visualized in an enrichment bar chart (Figure 3A, Appendix A).

KEGG pathway analysis identified 124 pathways, with notable enrichments observed in the MAPK signaling pathway, pathways in cancer, the AGE-RAGE signaling pathway in diabetic complications, and the prolactin signaling pathway. The top 20 pathways, selected based on *p*-value, are presented in bubble plots (Figure 3B, Appendix A).

Additionally, Mapper analysis in the KEGG database identified a total of 303 pathways, categorized as 59 in metabolism, 30 in environmental information processing, 20 in cellular processes, 81 in organismal systems, and 95 in human disease (Figure 3C, Appendix A). The top five pathways with the highest number of enriched genes were visualized using a String diagram based on the quantity of enriched genes per pathway (Figure 3D). Furthermore, Figure 3E illustrates the critical targets involved in the MAPK signaling pathway’s role in CMM.

### 3.7. MCODE Clustering Analysis

To further investigate the mechanisms by which resveratrol regulates CMM, we constructed a modular network using the MCODE algorithm to identify core therapeutic targets. Through topological network analysis, we discovered that key pathways involved include pathways in cancer, the AGE-RAGE signaling pathway in diabetic complications, type II diabetes mellitus, and the JAK-STAT signaling pathway (Figure 4A). To elucidate the relationships between targets and pathways, a subset was generated and displayed as a network (Figure 4B), allowing us to explore the potential functions of targets within different clusters. A comprehensive analysis was conducted using Metascape, which performed MCODE cluster analysis on 324 intersection genes, revealing 13 common target modules for resveratrol treatment of CMM (Figure 4C, Appendix A). This highlights the complexity of the biological pathways involved in resveratrol’s therapeutic effects on CMM.

Enrichment analysis revealed that most clusters were involved in various biological processes, cellular components, molecular functions, and KEGG signaling pathways. Within the protein–protein interaction (PPI) network, the top 20 node degree targets were identified. Notably, MCODE1 and MCODE4 showed a significant association with resveratrol. MCODE1 enriched four genes: STAT3, STAT5B, LEP, and AGT. This module showed enrichment in pathways including neuroactive ligand–receptor interaction, the JAK-STAT signaling pathway, and cytokine–cytokine receptor interaction (Table 4). MCODE4 enriched five core genes: EGFR, FGF2, FGFR1, INSR, and MAPK3. This module was enriched in pathways such as melanoma, the Rap1 signaling pathway, and the Ras signaling pathway (Table 5); these enriched key genes are shown in a diamond shape in Figure 4C.

### 3.8. Construction of the Target-Pathway Network and Identification of Core Targets

Based on the components involved in the top 20 pathways and their interconnections, we constructed a target-pathway network using Cytoscape 3.9.1, comprising 348 nodes and 792 edges (Figure 5). Nodes in the network represent targets, components, or pathways, while edges denote interactions. In Figure 5, the pink triangle symbolizes resveratrol components, orange circles denote pathways, green squares represent target genes intersecting with resveratrol and CMM, and the purple, blue, and green V-shapes denote DM, IS, and CHD, respectively.

The top 20 targets, ranked by degree within the constructed network, have been listed in Table 6. Topological analysis of the network was conducted using the CytoNCA plugin, sorting target genes by degree value. Intersection analysis of the top 20 targets identified earlier and those in the PPI network highlighted common targets, visualized in a Venn diagram (Figure 6). Ultimately, EGFR, STAT3, MAPK3, FGF2, FGFR1, and STAT5B emerged as six core target genes. These genes exhibit extensive interactions with others in potential pathways such as the MAPK signaling pathway, pathways in cancer, the AGE-RAGE signaling pathway in diabetic complications, the prolactin signaling pathway, and growth hormone synthesis, secretion, and action. This suggests their pivotal roles in the treatment of CMM.

### 3.9. Molecular Docking

The docking interactions between resveratrol and the six core proteins (EGFR, STAT3, MAPK3, FGF2, FGFR1, and STAT5) were validated, yielding the binding energies between the small-molecule compound and the respective large-molecule proteins. Lower binding energies typically indicate stronger interactions and higher affinity between the ligand and receptor. The results of these binding energies are summarized in Table 7, where all energies were found to be <−4 kcal/mol.

Resveratrol was specifically selected as the focus for visualizing molecular docking diagrams with EGFR, STAT3, MAPK3, FGF2, FGFR1, and STAT5 (Figure 7). In general, affinity values greater than −5 kcal/mol suggest no predicted binding, values less than −5 kcal/mol suggest moderate predicted binding, and values less than −7 kcal/mol suggest strong predicted binding [9]. Notably, MAPK3, EGFR, and resveratrol exhibited the strongest binding energies among the interactions analyzed.

### 3.10. Molecular Dynamics Simulations

Molecular dynamics simulations provide critical insights into the dynamic stability of receptor–ligand complexes under physiological conditions. Resveratrol was selected for simulations with MAPK3, EGFR, FGFR1, FGF2, STAT5, and STAT3 to assess binding stabilities considering binding affinity, complex interactions, and compound structural diversity. The stability of each system was evaluated using root mean square deviation (RMSD) and root mean square fluctuation (RMSF) over a 100 ns simulation period. A stable system typically exhibits RMSD values fluctuating under 0.2 nm.

Figure 8 illustrates RMSD curves for the six protein–ligand complexes, showing stable trajectories throughout the simulation. These complexes maintained consistently low fluctuation ranges, indicating stability over time. Specifically, the MAPK3–resveratrol complex remained stable from 0 to 100 ns, while stability in the EGFR–resveratrol complex was achieved after 60 ns.

RMSF analysis revealed higher values at the curves’ beginnings and ends, suggesting initial and final spatial protein instability during dynamics. Residues in direct contact with ligands (highlighted in green) exhibited lower RMSF values due to stabilizing interactions. Overall, resveratrol showed efficacy in maintaining structural stability across these target proteins amidst dynamic environmental changes. The MAPK3–resveratrol complex exhibited reduced residue fluctuation, indicating enhanced rigidity in these regions (Figure 9).

Resveratrol formed diverse interactions with target proteins, including hydrogen bonds, water bridges, and hydrophobic interactions. Hydrogen bonding, a robust non-covalent interaction, played a significant role. Over the 0–100 ns period, complexes formed varying numbers of hydrogen bonds: EGFR–resveratrol: 0–5; FGF2–resveratrol; 1–8; FGFR1–resveratrol: 1–9; MAPK3–resveratrol: 1–5; STAT3–resveratrol: 1–4; and STAT5–resveratrol: 1–4 (Figure 10). Key residues involved in these interactions within MAPK–resveratrol complexes included GLN_122, ASP_123, and GLU_50 (Appendix A).

Similarly, stable binding of resveratrol with MAPK, FGFR1, and FGF2 was evident from the low RMSF values of ligands (Appendix A). The radius of gyration (Rg) indicated tight binding and system constraint correlating with protein folding. Stable Rg values were observed in the FGFR1–resveratrol, FGF2–resveratrol, and MAPK3–resveratrol complexes (Figure 11).

### 3.11. Acquisition and Target Validation of CMM in GEO Dataset

The datasets IS (GSE140275), CHD (GSE12288), and DM (GSE29221) retrieved from the GEO database underwent differential analysis using the “limma” R package, yielding 42 potential target genes (Appendix A). Volcano plots for each dataset (GSE140275, GSE12288, GSE29221) depicted differentially expressed genes (DEGs), with up-regulated genes marked in red and down-regulated genes in green (Figure 12A,C,E). Heatmaps illustrated the top 50 DEGs across GSE140275, GSE12288, and GSE29221 (Figure 12B,D,F). Subsequently, 18 genes common to both drug target genes and GEO datasets were identified (Appendix A). These 18 genes were further intersected with 324 resveratrol-related genes and 42 CMM-related genes from traditional network pharmacology databases, resulting in the identification of 3 common target genes. Notably, FGF2 emerged as a pivotal target in the context of resveratrol’s impact on CMM, as delineated by network pharmacology (Figure 12G).

## 4. Discussion

CMM represents a prevalent chronic condition worldwide, significantly contributing to mortality and disability rates across populations. Individuals with CMM face heightened risks of complications from various cardiovascular and metabolic diseases [14]. Hence, identifying effective pharmacological treatments and elucidating underlying pathogenic mechanisms are crucial clinical imperatives. Natural products possessing pharmacological activities offer distinct advantages in alleviating clinical symptoms and in the prevention and treatment of CMM [15,16].

Resveratrol exhibits antioxidant and anti-inflammatory properties, contributing to improved glucose and lipid metabolism, enhanced cardiovascular function, and delayed aging processes [17]. Research indicates that resveratrol reduces oxidative stress, modulates the renin–angiotensin system via AMPK activation, enhances endothelial function through the Nrf2-II enzyme pathway [18], and lowers blood pressure [19]. Furthermore, it demonstrates efficacy in mitigating ischemia–reperfusion injury in ischemic heart disease [20]. Resveratrol also regulates lipid metabolism, suppresses inflammation, and diminishes oxidative stress, thereby beneficially influencing various stages of atherogenesis and the progression of atherosclerosis [21].

In this study, we employed network pharmacology to construct drug–target pathway topological networks, facilitating the prediction of drug targets and mechanisms of action. By identifying shared targets between resveratrol and CMM, we revealed potential therapeutic effects of resveratrol on this condition by targeting key nodes in the protein–protein interaction network of common targets. Additionally, we conducted GO and KEGG analyses, uncovering several biological processes associated with resveratrol in CMM, such as positive regulation of cell proliferation [22], gene expression [23], response to estradiol [24], positive transcriptional regulation of the RNA polymerase II promoter [25], and positive regulation of protein phosphorylation [22]. These processes predominantly occur in extracellular matrix, cell surface, myofibrillar membrane, and platelet α-granule lumen locations. Molecular functions identified include protein binding, growth factor activity, RNA polymerase II transcription factor activity, and ligand-activated sequence-specific DNA binding.

Furthermore, KEGG analysis highlighted enriched pathways including the MAPK signaling pathway [26], PI3K-Akt signaling pathway [27], calcium signaling pathway [28], Rap1 signaling pathway [25], Ras signaling pathway, and JAK-STAT signaling pathway [29]. Importantly, targets of resveratrol and CMM play pivotal roles in inflammatory responses, such as inhibiting pro-inflammatory cytokine production via the JAK-STAT signaling pathway, mitigating maladaptive hypertrophy in cardiac remodeling through the MAPK signaling pathway, and exerting anti-apoptotic effects via the PI3K-Akt signaling pathway [30,31].

The cardiovascular protective effect of resveratrol is primarily attributed to its ability to reduce oxidative stress, regulate inflammation, and improve cardiovascular risk factors [32]. Oxidative stress plays a crucial role in the pathogenesis of various cardiovascular diseases [33,34]. Resveratrol exerts a protective effect by scavenging free radicals, thereby safeguarding cells from oxidative damage [35]. The mitogen-activated protein kinase (MAPK) cascade is a critical signaling pathway that regulates numerous cellular processes, including proliferation, cell survival, and apoptosis, under both normal and pathological conditions such as oxidative stress [36]. Excessive activation of the MAPK signaling pathway due to oxidative stress is associated with pathological cardiac hypertrophy; however, resveratrol can inhibit ROS-mediated activation of MAPK/ERK1/2 [37]. Furthermore, resveratrol is known to inhibit the production of pro-inflammatory cytokines, highlighting its anti-inflammatory properties. Studies have shown that resveratrol down-regulates the expression of NF-κB p65 and p38 MAPK, while up-regulating the expression of SIRT1, thereby reducing vascular inflammatory damage and atherosclerosis [38]. Yang et al. found that resveratrol inhibits the production of COX-2/PGE2 induced by slow hormones by activating SIRT1, which can inhibit AP-1 (MAPK activation) and NF-κB transcription factor (acetylation) [39]. Gao et al. discovered that resveratrol can reduce cardiac dysfunction and fibrosis induced by diabetes, an effect associated with a reduction in the inflammatory response and the down-regulation of the AT1R-ERK/p38 MAPK signaling pathway [40].

To identify pivotal proteins and compounds, we constructed a compound–target pathway network. By intersecting this network with a protein–protein interaction (PPI) network focused on common targets, we identified six core targets: EGFR, STAT3, MAPK3, FGF2, STAT5B, and FGFR1. Subsequent molecular docking validated these targets, revealing binding energies consistently below −4 kcal/mol, indicating high affinity. Among these, MAPK3 and EGFR exhibited the strongest affinity. EGFR, known for its widespread presence in fibroblasts, notably interacts with STAT3, presenting a promising therapeutic avenue to mitigate CMM [41].

In this study, we employed molecular dynamics simulations to investigate the binding interactions of resveratrol with EGFR, STAT3, MAPK3, FGF2, STAT5B, and FGFR1 in depth. Our findings indicate that resveratrol exhibits strong binding affinity with MAPK3, suggesting its potential to modulate MAPK3 activity. Notably, MAPK3 forms stable hydrogen bonds with nearly all small-molecule docking ligands studied. Further exploration of these interactions could pave the way for targeted therapeutic strategies in CMM.

Our study also revealed an intersection between CMM datasets from the GEO database and resveratrol targets, highlighting FGF2 as a common target. This underscores the pivotal role of FGF2 in the pathogenesis of CMM. FGF2, a member of the fibroblast growth factor family [42], is widely expressed in cardiovascular tissues and plays critical roles in cardiomyocyte cell cycle regulation, cardiac development, disease pathogenesis, and repair mechanisms. Additionally, FGF2 is involved in various metabolic processes, including bile acid metabolism, fatty acid metabolism, and glucose metabolism [43]. Research has demonstrated FGF2’s significant involvement in atherosclerotic plaque formation [42] and its influence on vascular smooth muscle cell behavior [44]. Notably, FGF2 is recognized as a potent regulator of inflammation, capable of inducing pro-inflammatory cytokine expression in human aortic smooth muscle cells and promoting a shift from a contractile to a secretory phenotype [45]. Our findings suggest that targeting FGF2 through resveratrol may offer therapeutic potential in managing the inflammatory response associated with CMM.

SwissADME is a web-based tool used to calculate essential physicochemical, pharmacokinetic, and drug-related parameters of molecules, crucial for predicting their medicinal potential [46]. Pharmacokinetic and drug-like activity simulations performed by SwissADME aid in discovering active compounds with novel structures, thereby enhancing the likelihood of successful clinical drug development [47]. According to SwissADME simulations, resveratrol demonstrates favorable intestinal solubility and blood–brain barrier permeability, is metabolizable by the liver, and exhibits promising drug-like properties. Additionally, ADMETlab analysis indicates minimal toxicity associated with resveratrol.

Resveratrol is a compound found in various plants and exists in both cis- and trans-isomers, with trans-resveratrol being more biologically active and stable than the cis-isomer. Most of the reported health benefits are attributed to the trans form [48]. Due to its structure and interaction with biological pathways, trans-resveratrol is believed to cross cell membranes more efficiently and exert antioxidant and anti-inflammatory effects [49]. A growing number of studies have shown that trans-resveratrol has higher bioavailability compared to regular resveratrol, particularly in cardiovascular health [50]. For instance, Sung et al. found that resveratrol can effectively treat myocardial infarction induced by pressure overload by improving diastolic function, cardiac remodeling, myocardial energy, vascular function, and reducing cardiac fibrosis [51]. Additionally, Guo et al. discovered that resveratrol can treat atherosclerosis by inhibiting the TGF/ERK signaling pathway [52].

Due to the promising preclinical effects of resveratrol, many randomized clinical trials (RCTs) have reported similar findings [32,53,54]. However, the appropriate dose of resveratrol remains uncertain. Studies indicate that resveratrol provides health benefits in a dose-dependent manner, with low doses appearing to prevent various diseases [55]. Rapid metabolism in the intestine and liver results in low bioavailability and limited effectiveness, encouraging the exploration of methods to improve its bioavailability [56]. One approach is to combine resveratrol with other phytochemicals to protect it from rapid metabolism [57]. Alternatively, using resveratrol in different forms may enhance its efficacy [58]. Techniques such as derivatization, microencapsulation, nanomaterials, and bioenhancers have also been reported to be effective [59,60,61].

However, this study has some limitations. The reliability and accuracy of our compound and target predictions depend on the quality of the database data. While data mining methods provided valuable insights, validating these predictions through clinical trials and animal experiments would be of greater significance. Network pharmacology is primarily a data-driven and web-based research methodology; therefore, these findings should be further validated through clinical trials and animal experiments.

## 5. Conclusions

In conclusion, this study demonstrates the application of drug prediction techniques to explore phytochemicals for treating CMM. It systematically investigates the pharmacological and molecular mechanisms of resveratrol using multiple bioinformatics approaches, including network pharmacology, molecular docking, and molecular dynamics simulations. Resveratrol shows potential in treating CMM by mitigating pathological injury, inflammatory responses, and oxidative stress through diverse pathways. Molecular dynamics simulations have demonstrated the binding efficiencies of complexes such as MAPK3 and EGFR with resveratrol, highlighting its therapeutic potential through multi-target interactions. These findings provide valuable insights for the application and further development of resveratrol in managing CMM.

## Figures and Tables

**Figure 1 nutrients-16-02488-f001:**
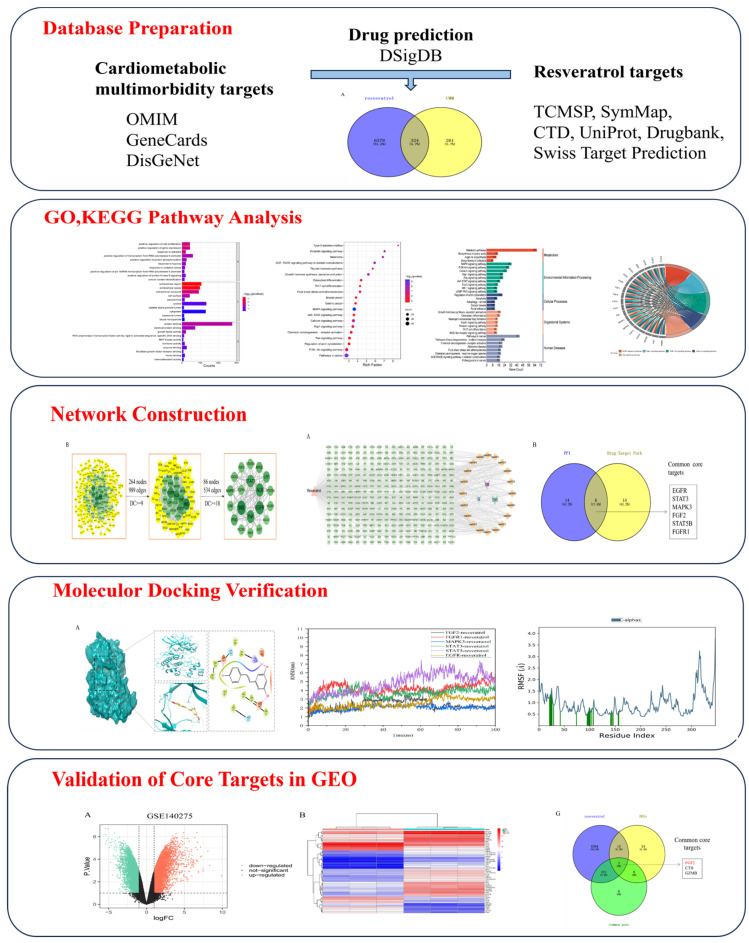
A detailed workflow of the network pharmacological investigation strategy for resveratrol in the treatment of CMM.

**Figure 2 nutrients-16-02488-f002:**
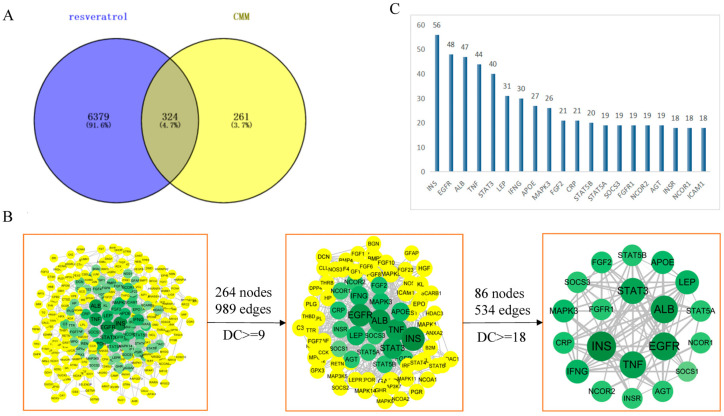
Screening of resveratrol in CMM. (**A**) Venn diagram of intersecting target genes of resveratrol and CMM. (**B**) Protein–protein interaction network. (**C**) The top 20 proteins in the PPI network in terms of degree values.

**Figure 3 nutrients-16-02488-f003:**
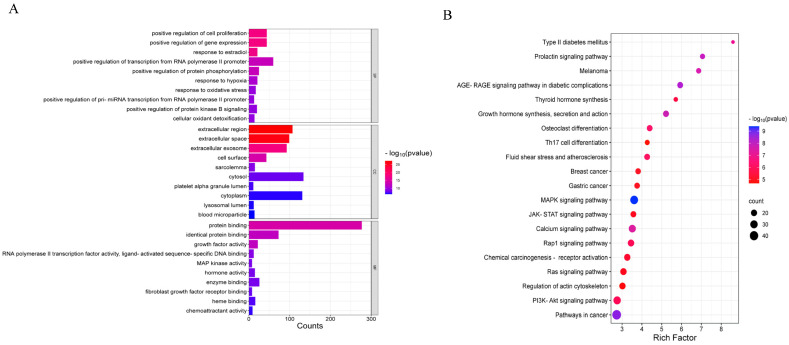
GO and KEGG enrichment analysis. (**A**) Bar chart of GO enrichment analysis. (**B**) Bubble plot of KEGG enrichment. (**C**) Results of KEGG enrichment analysis (different colors indicate different systems of action, and pathways in each system are arranged in descending order of the number of enriched genes). (**D**) KEGG key pathway network. (**E**) MAPK signaling pathway.

**Figure 4 nutrients-16-02488-f004:**
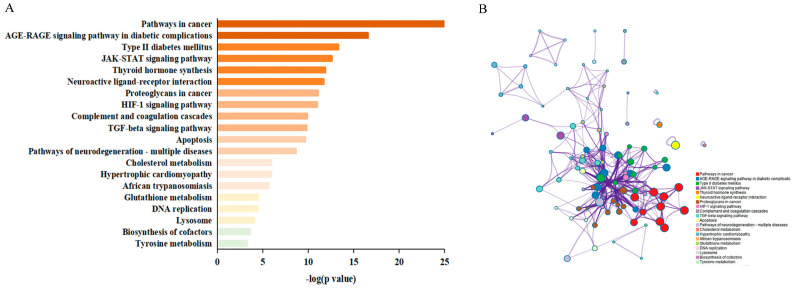
Cluster module analysis diagram of related protein targets of resveratrol in CMM. (**A**) Highly enriched terms of resveratrol in CMM. (**B**) Sub network specific to the interaction. (**C**) Cluster analysis of resveratrol in CMM.

**Figure 5 nutrients-16-02488-f005:**
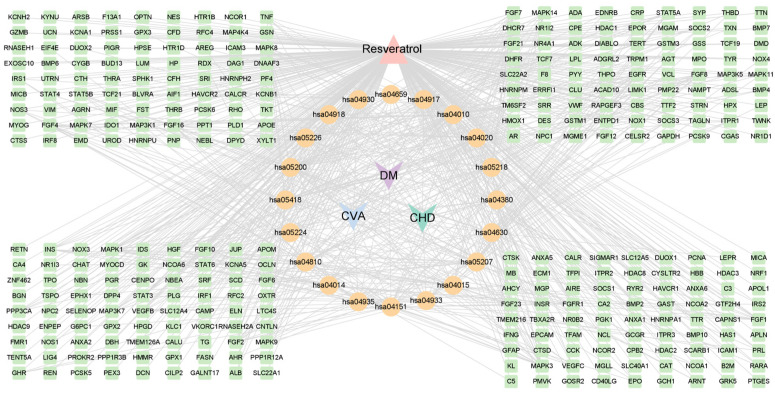
Component–target pathway diagram of resveratrol in CMM.

**Figure 6 nutrients-16-02488-f006:**
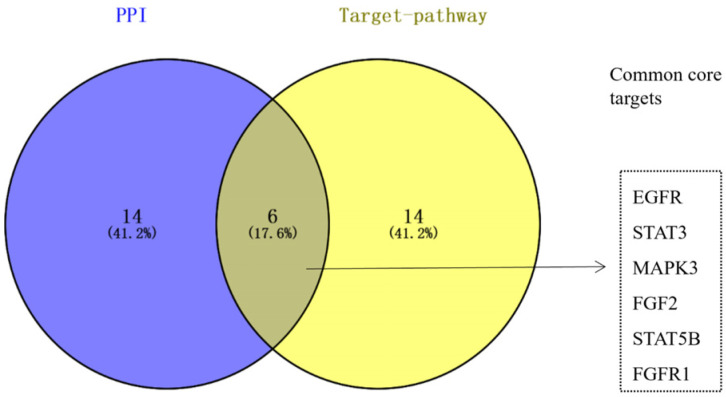
Identification of core targets from resveratrol–PPI and drug–target path analysis.

**Figure 7 nutrients-16-02488-f007:**
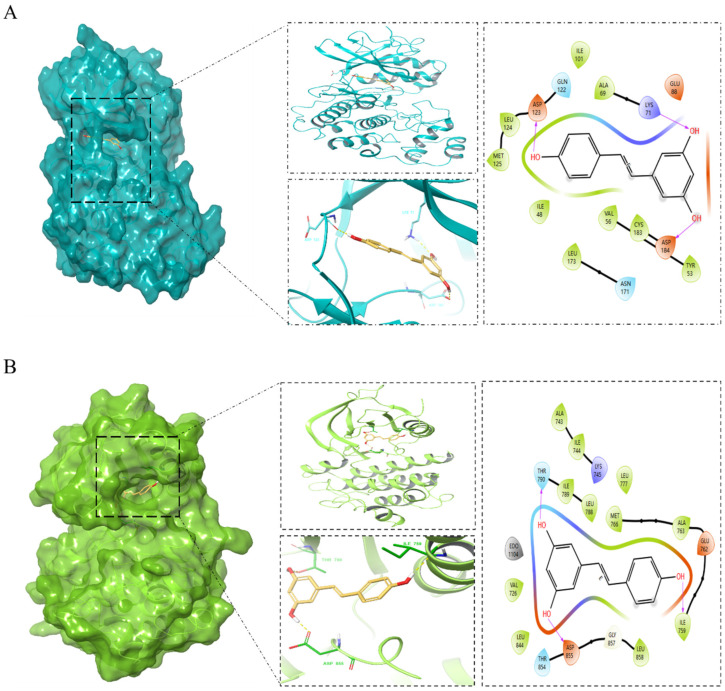
Molecular docking results in each target with resveratrol. (**A**) MAKP3, (**B**) EGFR, (**C**) FGFR1, (**D**) FGF2, (**E**) STAT5, and (**F**) STAT3.

**Figure 8 nutrients-16-02488-f008:**
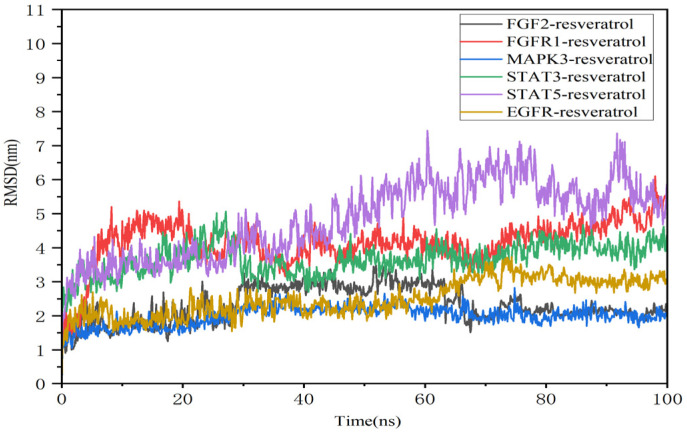
The fluctuation plot of the target protein–ligand complexes’ RMSD values.

**Figure 9 nutrients-16-02488-f009:**
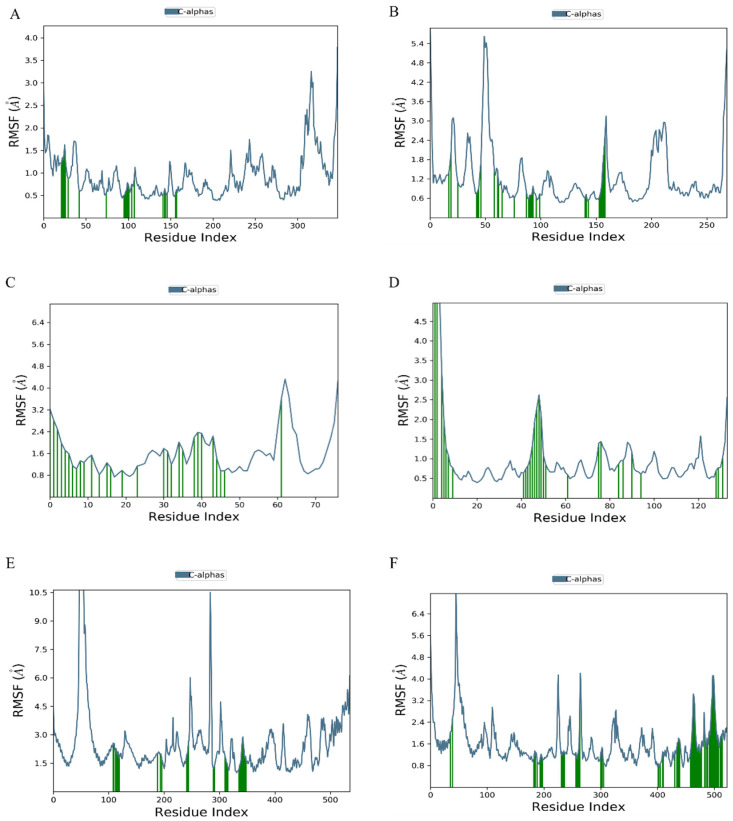
The fluctuation plot of the target protein–ligand complexes’ RMSF values. (**A**) MAKP3, (**B**) EGFR, (**C**) FGFR1, (**D**) FGF2, (**E**) STAT5, and (**F**) STAT3. Residues in contact with the ligand are marked in green.

**Figure 10 nutrients-16-02488-f010:**
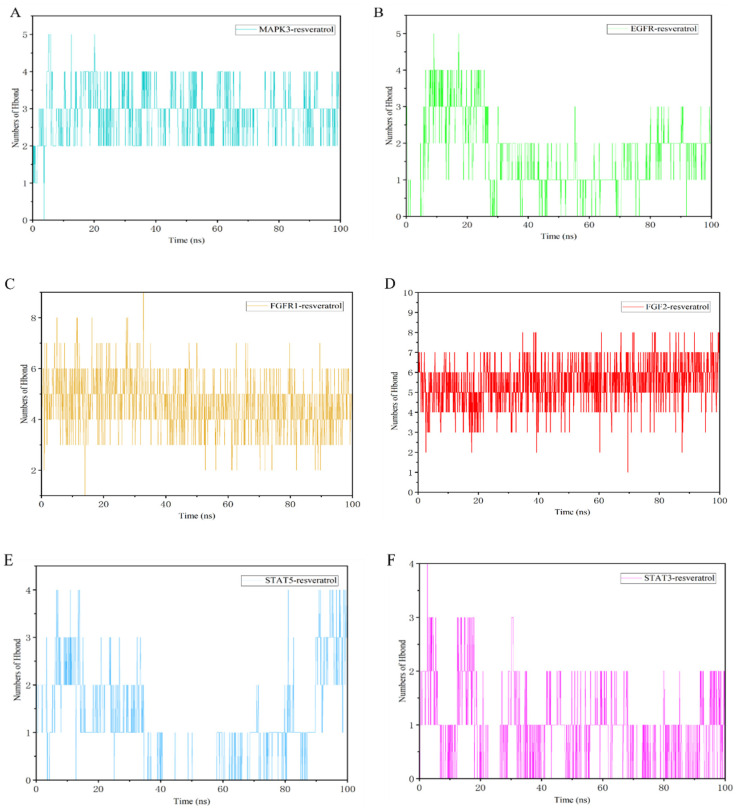
Hydrogen bond number of complexes. (**A**) MAKP3–resveratrol, (**B**) EGFR–resveratrol, (**C**) FGFR1–resveratrol, (**D**) FGF2–resveratrol, (**E**) STAT5–resveratrol, and (**F**) STAT3–resveratrol.

**Figure 11 nutrients-16-02488-f011:**
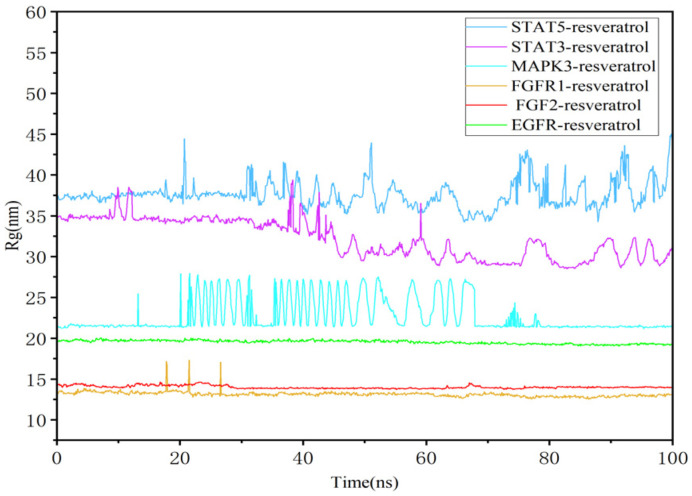
Compactness of the protein according to Rg.

**Figure 12 nutrients-16-02488-f012:**
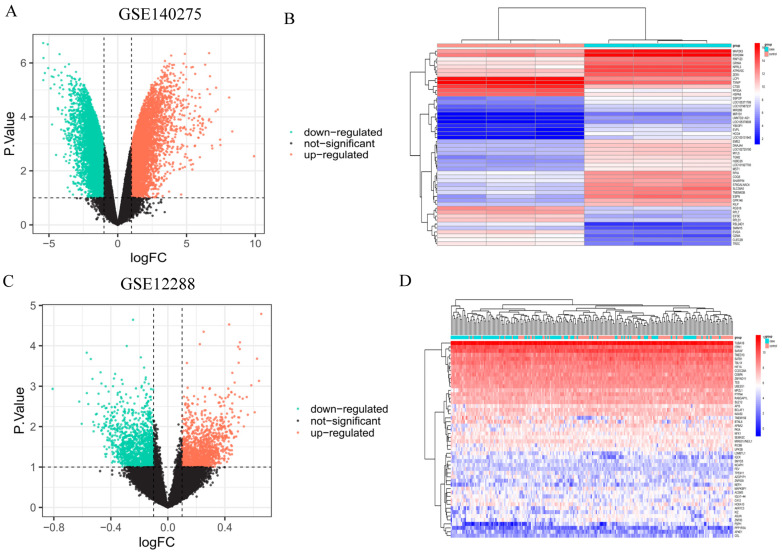
Validation of core resveratrol targets using GEO datasets. (**A**) Volcano plot of the IS dataset GSE140275. (**B**) Volcano plot of the CHD dataset GSE12288. (**C**) Volcano plot of the DM dataset GSE29221. (**D**) Heatmap of differential genes in the IS dataset GSE140275. (**E**) Heatmap of differential genes in the CHD dataset GSE12288. (**F**) Heatmap of differential genes in the DM dataset GSE29221. (**G**) Validation of resveratrol core target genes using the GEO dataset.

**Table 1 nutrients-16-02488-t001:** Drug prediction of CMM.

Name	*p* Value	Combined Score
D-Norepinephrine d-bitartrate	3.37 × 10^−29^	747.3471
Liothyronine	2.24 × 10^−26^	625.8880
UNII-CXY7B3Q98Z	1.38 × 10^−24^	534.0072
Corticosterone	5.17 × 10^−24^	540.7950
Resveratrol	9.78 × 10^−24^	170.8037
Estradiol	1.89 × 10^−23^	127.9177
Tamoxifen	2.11 × 10^−23^	220.5620
Cholesterol	4.47 × 10^−23^	484.1772
4-aminobutyric acid	5.19 × 10^−23^	500.0786
Arsenenous acid	9.13 × 10^−23^	173.6427

**Table 2 nutrients-16-02488-t002:** Drug similarity analysis of resveratrol.

Compound	Lipinski Rules
Molecular Weight (MW)	Hydrogen Bonding Acceptor (HBA)	Hydrogen Bonding Donor (HBD)	Moriguchi Octanol–Water Partition Coefficient (MLogP)	Topological Surface Area (Å^2^)
Resveratrol	<500	<10	≤5	≤4.15	<140
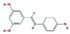	228.24	3	3	2.26	60.69

**Table 3 nutrients-16-02488-t003:** Toxicity analysis of resveratrol.

Parameter	Resveratrol
hERG Blockers	Non-blocker
Human hepatotoxicity	Negative
Eye corrosion	Negative
Respiratory toxicity	Negative
Rat oral acute toxicity	Negative

**Table 4 nutrients-16-02488-t004:** Pathway enrichment results for MCODE1.

Color	MCODE	Go	Description	Log10(P)
	MCODE1	hsa04080	Neuroactive ligand–receptor interaction	−23.2
	MCODE1	hsa04630	JAK-STAT signaling pathway	−8.6
	MCODE1	hsa04060	Cytokine–cytokine receptor interaction	−5.6

**Table 5 nutrients-16-02488-t005:** Pathway enrichment results for MCODE4.

Color	MCODE	Go	Description	Log10(P)
	MCODE4	hsa05218	Melanoma	−32.6
	MCODE4	hsa04015	Rap1 signaling pathway	−29.2
	MCODE4	hsa04014	Ras signaling pathway	−28.5

**Table 6 nutrients-16-02488-t006:** Top 20 targets, ranked by degree in the compound–target pathway network of resveratrol in CMM.

Gene name	Degree	Gene Name	Degree	Gene Name	Degree	Gene Name	Degree
MAPK1	17	FGF6	12	FGF1	11	FGFR1	10
MAPK3	17	FGF7	12	MAPK9	11	HGF	9
EGFR	13	FGF8	12	MAPK8	11	MAPK14	9
FGF2	12	FGF16	12	FGF23	11	STAT3	8
FGF4	12	FGF10	12	FGF21	11	STAT5B	8

**Table 7 nutrients-16-02488-t007:** Resveratrol molecular docking scores.

Gene	MAPK3	EGFR	FGFR1	FGF2	STAT5	STAT3
Binding energy (kcal/mol)	−7.817	−7.496	−6.254	−4.402	−4.395	−4.390

## Data Availability

The original contributions presented in the study are included in the article/Appendix A.

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
