# Peer review of "Investigating the Molecular Mechanisms of Resveratrol in Treating Cardiometabolic Multimorbidity: A Network Pharmacology and Bioinformatics Approach with Molecular Docking Validation"

_nutrients, 2024, doi:10.3390/nu16152488_

Round 1

Reviewer 1 Report

Comments and Suggestions for Authors

The study presented by Gong and colleagues is interesting. In this paper, the authors utilize various bioinformatics databases to predict the potential effects of resveratrol on cardiometabolic diseases. To achieve their final objective, they primarily employ databases commonly used in omics sciences studies. Furthermore, the authors predict the interaction of resveratrol with selected target proteins. While the study has limitations, as acknowledged by the authors, I believe it could be of great interest to the scientific community. However, it requires improvements in its presentation.

In section 3.7, the authors indicate that MCODE4 is enriched in 5 genes and MCODE1 in 4 genes, referring to tables 4 and 5, where 3 and 4 are listed, respectively. The authors should clarify this section.

There are errors in the order of the references, reference 2 should be cited before reference 3.

On page 11, the caption of figure 4 should be placed below the figure, not in the middle of the text.

Tables 4 and 5 lack captions that clarify the sections of the table.

Figure 5 is cut off and needs revision

Author Response

Comments 1: In section 3.7, the authors indicate that MCODE4 is enriched in 5 genes and MCODE1 in 4 genes, referring to tables 4 and 5, where 3 and 4 are listed, respectively. The authors should clarify this section.

Response 1: Thank you for your suggestion, we have explained the reasons in the methods section on the page of 11, line 6-23, and highlighted in red in the text.

To further investigate the mechanisms by which resveratrol regulates CMM, we constructed a modular network using the MCODE algorithm to identify core therapeutic targets. Through topological network analysis, we discovered that key pathways involved include pathways in cancer, the AGE-RAGE signaling pathway in diabetic complications, Type II diabetes mellitus, and the JAK-STAT signaling pathway (Fig.4A). To elucidate the relationships between targets and pathways, a subset was generated and displayed as a network (Fig.4B), allowing us to explore the potential functions of targets within different clusters. A comprehensive analysis was conducted using Metascape, which performed MCODE cluster analysis on 324 intersection genes, revealing 13 common target modules for resveratrol treatment of CMM (Fig.4C, Supplementary Table S16). This highlights the complexity of the biological pathways involved in resveratrol's therapeutic effects on CMM.

Enrichment analysis revealed that most clusters were involved in various biological processes, cellular components, molecular functions, and KEGG signaling pathways (Supplementary Table S16). Within the protein-protein interaction (PPI) network, the top 20 node degree targets were identified. Notably, MCODE1 and MCODE4 showed a significant association with resveratrol. MCODE1 enriched four genes: STAT3, STAT5B, LEP, and AGT. This module showed enrichment in pathways including Neuroactive ligand-receptor interaction, the JAK-STAT signaling pathway, and Cytokine-cytokine receptor interaction (Table 4). MCODE4 enriched five core genes: EGFR, FGF2, FGFR1, INSR, and MAPK3. This module was enriched in pathways such as Melanoma, the Rap1 signaling pathway, and the Ras signaling pathway (Table 5), these enriched key genes are shown in a diamond shape in Fig.4C.

Figure 4. Cluster module analysis diagram of related protein targets of resveratrol in CMM. (A) Highly enriched terms of resveratrol in CMM. (B) Sub network specific to the interaction. (C) Cluster analysis of resveratrol in CMM.

Comments 2: There are errors in the order of the references, reference 2 should be cited before reference 3.

Response 2: Thank you for your suggestion, we have revised the order of references on the page of 2 line 6-9 in the text.

Comments 3: On page 11, the caption of figure 4 should be placed below the figure, not in the middle of the text.

Response 3: Thank you for your suggestion, we have revised the position of the caption in Figure 4 on the page 11 in the text.

Comments 4: Tables 4 and 5 lack captions that clarify the sections of the table.

Response 4: Thank you for your suggestion, we have revised the Tables 4 and 5 and modified it on the page of 11 line 4-5 in the text.

Comments 5: Figure 5 is cut off and needs revision

Response 5: Thank you for your suggestion, we have revised the Figure 5 on the page of 12 in the text.

Reviewer 2 Report

Comments and Suggestions for Authors

This study analysed the pharmacological and molecular mechanisms of resveratrol through techniques such as network pharmacology, molecular docking and molecular dynamics simulations. Resveratrol was found to act on chronic granulocytic leukemia and osteomyelitis through multiple signaling pain and metabolic pathways. Similarly, the present study analyzed in detail the gene level changes and association of important proteins due to resveratrol. However, the logic of cardiovascular metabolic diseases suppressed by resveratrol is unclear and lacks comparison and description of the metabolic pathways involved. Further modifications and additions to the correlation are needed.
1. Resveratrol exists in trimeric, dimeric and other structures, as well as cis and trans structures. Which structure of resveratrol studied in this study should be clarified in the manuscript. Do the structural differences between cis- and trans-resveratrol have some impact on the analysis of molecular docking and molecular dynamics?
2. Please give a clearer version of Figure 3E.
3. In 3.7, this study analyzed the association of resveratrol on genes associated with chronic osteoarthritis. May I ask why genes associated with osteoarthritis were analyzed in this study when the focus was on cardiac diseases? Is there any correlation?
4. Twenty relevant core proteins were analysed in this study, but the results of molecular docking of resveratrol with six core proteins are shown in Table 2. May I ask if the remaining 14 core proteins were analysed and why only these 6 proteins were analysed?
5. It is recommended that a description of resveratrol's role in cardiovascular disease through the MAPK signalling pathway be added to the manuscript.
6. This study mentions cardiovascular metabolic diseases and chronic granulocytic leukaemia in the preface section and chronic osteomyelitis in the manuscript. Please clarify the relevance of these three diseases in the manuscript.

Comments on the Quality of English Language

Minor editing of English language required

Author Response

Comments 1: Resveratrol exists in trimeric, dimeric and other structures, as well as cis and trans structures. Which structure of resveratrol studied in this study should be clarified in the manuscript. Do the structural differences between cis- and trans-resveratrol have some impact on the analysis of molecular docking and molecular dynamics?
Response 1: Thank you for your suggestion, we have added the structure of resveratrol on the page of 7, line 19-20, on the page of 20, line 42-47, and highlighted in red in the text.

Resveratrol is a compound found in various plants. Due to the presence of a central ethylene moiety in its structure, resveratrol exists in two natural isomers: cis- and trans-resveratrol. Cis-resveratrol is the Z geometric isomer, while trans-resveratrol is the E geometric isomer. Naturally occurring resveratrol is predominantly found in the trans E-configuration[1]. When exposed to ultraviolet and visible light, trans-resveratrol photoisomerizes into cis-resveratrol, which is less stable and not commercially available [2,3]. Cis-resveratrol is less potent and has less data available regarding its effects on human health compared to trans-resveratrol [4].

Resveratrol activates and regulates NAD+_ dependent enzymes known as sirtuins (SIRT). According to Grathwol et al., the E-isomer (trans-resveratrol) has a favorable inhibition effect on SIRT2 from a thermodynamic standpoint. Conversely, the Z-isomer (cis-resveratrol) is a metastable form, and its biological efficacy is reduced upon photoisomerization from trans-resveratrol due to ultraviolet radiation exposure [5].

Trans-resveratrol is more prevalent and has been associated with various biological activities, such as inducing cell cycle arrest, differentiation, and apoptosis [6-8]. Therefore, trans-resveratrol is more biologically active and stable than its cis counterpart [9], with most reported health benefits attributed to this isomer [10,11]. To ensure effective binding, we consider only the binding of individual molecules to proteins.

This study focuses on the effects of trans-resveratrol on cardiometabolic multimorbidity.

References

[1]          Stervbo, U., O. Vang, and C. Bonnesen, A review of the content of the putative chemopreventive phytoalexin resveratrol in red wine. Food Chemistry, 2007. 101(2): p. 449-457.

[2]          Eric Wei Chiang Chan, C.W.W., Yong Hui Tan, Jenny Pei Yan Foo, Siu Kuin Wong, Hung Tuck Chan, Resveratrol and pterostilbene: A comparative overview of their chemistry, biosynthesis, plant sources and pharmacological properties. Vol. Volume: 9. 2019: Issue: 7. 124-129.

[3]          Intagliata, S., et al., Strategies to Improve Resveratrol Systemic and Topical Bioavailability: An Update. Antioxidants (Basel), 2019. 8(8).

[4]          Orgován, G., I. Gonda, and B. Noszál, Biorelevant physicochemical profiling of (E)- and (Z)-resveratrol determined from isomeric mixtures. J Pharm Biomed Anal, 2017. 138: p. 322-329.

[5]          Grathwol, C.W., et al., Activation of Sirtuin 2 Inhibitors Employing Photoswitchable Geometry and Aqueous Solubility. ChemMedChem, 2020. 15(15): p. 1480-1489.

[6]          Akinwumi, B.C., K.M. Bordun, and H.D. Anderson, Biological Activities of Stilbenoids. Int J Mol Sci, 2018. 19(3).

[7]          Anisimova, N.Y., et al., Trans-, cis-, and dihydro-resveratrol: a comparative study. Chem Cent J, 2011. 5: p. 88.

[8]          Orallo, F., Comparative studies of the antioxidant effects of cis- and trans-resveratrol. Curr Med Chem, 2006. 13(1): p. 87-98.

[9]          Moreno, A., M. Castro, and E. Falqué, Evolution of trans- and cis-resveratrol content in red grapes (Vitis vinifera L. cv Mencía, Albarello and Merenzao) during ripening. European Food Research and Technology, 2008. 227(3): p. 667-674.

[10]        Riccio, B.V.F., et al., Resveratrol isoforms and conjugates: A review from biosynthesis in plants to elimination from the human body. Arch Pharm (Weinheim), 2020. 353(12): p. e2000146.

[11]        Mukherjee, S., J.I. Dudley, and D.K. Das, Dose-dependency of resveratrol in providing health benefits. Dose Response, 2010. 8(4): p. 478-500.

Comments 2: Please give a clearer version of Figure 3E.

Response 2: Thank you for your suggestion, we have modified the Figure 3E. on the page of 12 in the text.

Comments 3: In 3.7, this study analyzed the association of resveratrol on genes associated with chronic osteoarthritis. May I ask why genes associated with osteoarthritis were analyzed in this study when the focus was on cardiac diseases? Is there any correlation?

Response 3: Thank you for your suggestion.

The disease studied in this paper is cardiometabolic multimorbidity, abbreviated as CMM. Since the introduction of this paper, the short form of cardiometabolic multimorbidity (CMM) has been adopted, and chronic osteoarthritis is not involved in this paper. The topic you mentioned is also very interesting and valuable, we can further analyze and discuss it in the future.

Comments 4: Twenty relevant core proteins were analysed in this study, but the results of molecular docking of resveratrol with six core proteins are shown in Table 2. May I ask if the remaining 14 core proteins were analysed and why only these 6 proteins were analysed?

Response 4: Thank you for your suggestion, we have added some details on the page of 12, line 1-5, Figure 5, Table 6 and highlighted in red in the text.

Using Cytoscape software, compounds, KEGG pathways, and their corresponding targets were combined to construct a "compound-target-pathway" network (Fig.5). Table 6 lists the top 20 targets sorted by degree value in the compound-target-pathway network. A Venn diagram was used to identify common targets between the top 20 targets of the "compound-disease" PPI network and the "compound-target-pathway" network. Six common core proteins were identified (Fig.6), while the remaining 14 proteins were not core proteins. These six core proteins were subsequently selected for molecular docking and molecular dynamics simulations.

Figure 5. Component-target-pathway diagram of resveratrol in CMM.

Gene name

Degree

Gene name

Degree

Gene name

Degree

Gene name

Degree

MAPK1

17

FGF6

12

FGF1

11

FGFR1

10

MAPK3

17

FGF7

12

MAPK9

11

HGF

9

EGFR

13

FGF8

12

MAPK8

11

MAPK14

9

FGF2

12

FGF16

12

FGF23

11

STAT3

8

FGF4

12

FGF10

12

FGF21

11

STAT5B

8

Table 6. Top 20 targets, ranked by degree in the compound-target-pathway network of resveratrol in CMM.

Figure 6. Identification of core targets from resveratrol-PPI and Drug-Target-Path analysis.

This approach allows for a clearer understanding of the complex interactions between "resveratrol- cardiometabolic multimorbidity" and the "resveratrol-target-pathway network," enabling more accurate identification of the most critical proteins[12].

References

[12]        Yu, X., et al., Analyzing the molecular mechanism of xuefuzhuyu decoction in the treatment of pulmonary hypertension with network pharmacology and bioinformatics and verifying molecular docking. Computers in Biology and Medicine, 2024. 169: p. 107863.

Comments 5: It is recommended that a description of resveratrol's role in cardiovascular disease through the MAPK signaling pathway be added to the manuscript.
Response 5: Thank you for your suggestion, we have modified the unit on the page of 20, line 4-15, and highlighted in red in the text.

The cardiovascular protective effect of resveratrol is primarily attributed to its ability to reduce oxidative stress, regulate inflammation, and improve cardiovascular risk factors [13]. Oxidative stress plays a crucial role in the pathogenesis of various cardiovascular diseases [14-15]. Resveratrol exerts a protective effect by scavenging free radicals, thereby safeguarding cells from oxidative damage [16]. The mitogen-activated protein kinase (MAPK) cascade is a critical signaling pathway that regulates numerous cellular processes, including proliferation, cell survival, and apoptosis, under both normal and pathological conditions such as oxidative stress [17]. Excessive activation of the MAPK signaling pathway due to oxidative stress is associated with pathological cardiac hypertrophy; however, resveratrol can inhibit ROS-mediated activation of MAPK/ERK1/2 [18]. Furthermore, resveratrol is known to inhibit the production of pro-inflammatory cytokines, highlighting its anti-inflammatory properties. Studies have shown that resveratrol down-regulates the expression of NF-κB p65 and p38 MAPK, while up-regulating the expression of SIRT1, thereby reducing vascular inflammatory damage and atherosclerosis [19]. Yang et al. found that resveratrol inhibits the production of COX-2/PGE2 induced by slow hormones by activating SIRT1, which can inhibit AP-1 (MAPKs activation) and NF-κB transcription factor (acetylation) [20]. Gao et al. discovered that resveratrol can reduce cardiac dysfunction and fibrosis induced by diabetes, an effect associated with the reduction of inflammatory response and the downregulation of the AT1R-ERK/p38 MAPK signaling pathway [21].

References

[13]   Gal, R., et al., The Effect of Resveratrol on the Cardiovascular System from Molecular Mechanisms to Clinical Results. Int J Mol Sci, 2021. 22(18).

[14]   Tsutsui, H., S. Kinugawa, and S. Matsushima, Oxidative stress and heart failure. Am J Physiol Heart Circ Physiol, 2011. 301(6): p. H2181-90.

[15]   Oyewole, A.O. and M.A. Birch-Machin, Mitochondria-targeted antioxidants. Faseb j, 2015. 29(12): p. 4766-71.

[16]    Malaguarnera, L., Influence of Resveratrol on the Immune Response. Nutrients, 2019. 11(5).

[17]        Muslin, A.J., MAPK signalling in cardiovascular health and disease: molecular mechanisms and therapeutic targets. Clin Sci (Lond), 2008. 115(7): p. 203-18.

[18]    Singh, A.K. and M. Vinayak, Resveratrol alleviates inflammatory hyperalgesia by modulation of reactive oxygen species (ROS), antioxidant enzymes and ERK activation. Inflamm Res, 2017. 66(10): p. 911-921.

[19]        Deng, Z.Y., et al., Resveratrol alleviates vascular inflammatory injury by inhibiting inflammasome activation in rats with hypercholesterolemia and vitamin D2 treatment. Inflamm Res, 2015. 64(5): p. 321-32.

[20]        Yang, C.M., et al., Resveratrol inhibits BK-induced COX-2 transcription by suppressing acetylation of AP-1 and NF-κB in human rheumatoid arthritis synovial fibroblasts. Biochem Pharmacol, 2017. 132: p. 77-91.

[21]        Gao, Y., et al., Resveratrol Ameliorates Diabetes-Induced Cardiac Dysfunction Through AT1R-ERK/p38 MAPK Signaling Pathway. Cardiovasc Toxicol, 2016. 16(2): p. 130-7.

Comments 6: This study mentions cardiovascular metabolic diseases and chronic granulocytic leukaemia in the preface section and chronic osteomyelitis in the manuscript. Please clarify the relevance of these three diseases in the manuscript.

Response 6: Thank you for your suggestion.

The disease studied in this paper is cardiometabolic multimorbidity, abbreviated as CMM. Since the introduction of this article, the short form of cardiometabolic multimorbidity (CMM) has been adopted. chronic granulocytic leukaemia and chronic osteomyelitis were not mentioned in the article. The topic you mentioned is also very valuable, we further analyze and discuss it in the future.

Reviewer 3 Report

Comments and Suggestions for Authors

The article affords a comprehensive modeling investigation of the molecular mechanisms of resveratrol towards targets and mechanisms involved in cardiometabolic multimorbidity.

The subject is very interesting and urgent based both on the widepsread incidence and social relevance of the considered condition, and the long overdue clarification of the potential of resveratrol. I have particularly appreciated, among other things, the clarification of the compliance with the Lipinski Rules.

In the frame of the scope and aims of the study, the analysis is comprehensive and I have no concerns. Only in Section 3.9 Molecular docking (and elsewhere), Authors should use either kJ/mol or kcal/mol and not both such units.

However, under the limitations of the study, at the end of the Discussion section, it would be useful to expand the suggestions for further research, especially considering that existing food supplements are known to contain far lower amounts of resveratrol than the observed effective dose in animal studies or clinical trials. Thus, the topics of effective dose and bioavailability/bioaccessibility should be stressed as suggestions for further research.

Moreover, resveratrol is usually presented on the market in the form of trans-resveratrol, and some words also on this topic would be useful, for example whether trans-resveratrol is expected to provide similar, worse or better functions compared to resveratrol in the frame of this study.

All the above stated will be quite useful to address the field of extraction of natural products rich in resveratrol, and the often challenging purification issue.

Author Response

Comments 1: In the frame of the scope and aims of the study, the analysis is comprehensive and l have no concerns. Only in Section 3.9 Molecular docking (and elsewhere), Authors should use either kJ/mol or kcal/mol and not both such units.

Response 1: Thank you for your suggestion, we have modified the unit on the page of 13, line 6-9, Table 7 and highlighted in red in the text.

Comments 2: However, under the limitations of the study, at the end of the Discussion section, it would be useful to expand the suggestions for further research, especially considering that existing food supplements are known to contain far lower amounts of resveratrol than the observed effective dose in animal studies or clinical trials. Thus, the topics of effective dose and bioavailability/bio-accessibility should be stressed as suggestions for further research.

Response 2: Thank you for your suggestion, we have given more in-depth suggestions on the topic of effective dosage and bioavailability/bioaccessibility of resveratrol to emphasize this point on the page of 21, line 11-20, and highlighted in red in the text.

Due to the promising preclinical effects of resveratrol, many randomized clinical trials (RCTs) have reported similar findings[1-3]. However, the appropriate dose of resveratrol remains uncertain. Studies indicate that resveratrol provides health benefits in a dose-dependent manner, with low doses appearing to prevent various diseases[4]. Rapid metabolism in the intestine and liver results in low bioavailability and limited effectiveness, encouraging the exploration of methods to improve its bioavailability[5]. One approach is to combine resveratrol with other phytochemicals to protect it from rapid metabolism[6]. Alternatively, using resveratrol in different forms may enhance its efficacy[7]. Techniques such as derivatization, microencapsulation, nanomaterials, and bioenhancers have also been reported to be effective[8-10].

References

[1]          Teimouri, M., et al., Anti-inflammatory effects of resveratrol in patients with cardiovascular disease: A systematic review and meta-analysis of randomized controlled trials. Complement Ther Med, 2022. 70: p. 102863.

[2]          Raj, P., et al., A Comprehensive Analysis of the Efficacy of Resveratrol in Atherosclerotic Cardiovascular Disease, Myocardial Infarction and Heart Failure. Molecules, 2021. 26(21):p.6600.

[3]          Gal, R., et al., The Effect of Resveratrol on the Cardiovascular System from Molecular Mechanisms to Clinical Results. Int J Mol Sci, 2021. 22(18):p.10152.

[4]          Brown, K., et al., Resveratrol for the Management of Human Health: How Far Have We Come? A Systematic Review of Resveratrol Clinical Trials to Highlight Gaps and Opportunities. Int J Mol Sci, 2024. 25(2):p.747.

[5]          Prakash, V., et al., Resveratrol as a Promising Nutraceutical: Implications in Gut Microbiota Modulation, Inflammatory Disorders, and Colorectal Cancer. International Journal of Molecular Sciences, 2024. 25(6): p. 3370.

[6]          Wightman, E.L., et al., Effects of resveratrol alone or in combination with piperine on cerebral blood flow parameters and cognitive performance in human subjects: a randomised, double-blind, placebo-controlled, cross-over investigation. Br J Nutr, 2014. 112(2): p. 203-13.

[7]          Trotta, V., et al., In vitro biological activity of resveratrol using a novel inhalable resveratrol spray-dried formulation. Int J Pharm, 2015. 491(1-2): p. 190-7.

[8]          Sandhir, R., N. Singhal, and P. Garg, Chapter 13 - Increasing resveratrol bioavailability: A therapeutic challenge focusing on the mitochondria, in Mitochondrial Dysfunction and Nanotherapeutics, M.R. de Oliveira, Editor. 2021, Academic Press. p. 349-384.

[9]          Salla, M., et al., Enhancing the Bioavailability of Resveratrol: Combine It, Derivatize It, or Encapsulate It? Pharmaceutics, 2024. 16(4): p. 569.

[10]        de Vries, K., M. Strydom, and V. Steenkamp, A Brief Updated Review of Advances to Enhance Resveratrol's Bioavailability. Molecules, 2021. 26(14):p.4367.

Comments 3: Moreover, resveratrol is usually presented on the market in the form of trans-resveratrol, and some words also on this topic would be useful, for example whether trans-resveratrol is expected to provide similar, worse or better functions compared to resveratrol in the frame of this study.

Response 3: Thank you for your suggestion, we have made further additions about the topic of trans-resveratrol, second to last line on page 20, on the page of 21, line 1-10, and highlighted in red in the text.

Resveratrol is a compound found in various plants and exists in both cis- and trans-isomers, with trans-resveratrol being more biologically active and stable than the cis-isomer. Most of the reported health benefits are attributed to the trans form[11]. Due to its structure and interaction with biological pathways, trans-resveratrol is believed to cross cell membranes more efficiently and exert antioxidant and anti-inflammatory effects[12]. This study focuses on trans-resveratrol. A growing number of studies have shown that trans-resveratrol has higher bioavailability compared to regular resveratrol, particularly in cardiovascular health[13]. For instance, Sung et al. found that resveratrol can effectively treat myocardial infarction induced by pressure overload by improving diastolic function, cardiac remodeling, myocardial energy, vascular function, and reducing cardiac fibrosis[14]. Additionally, Guo et al. discovered that resveratrol can treat atherosclerosis by inhibiting the TGF/ERK signaling pathway[15].

References

[11]        Mukherjee, S., J.I. Dudley, and D.K. Das, Dose-dependency of resveratrol in providing health benefits. Dose Response, 2010. 8(4): p. 478-500.

[12]        Fiod Riccio, B.V., et al., Characteristics, Biological Properties and Analytical Methods of Trans-Resveratrol: A Review. Crit Rev Anal Chem, 2020. 50(4): p. 339-358.

[13]        Vikal, A., et al., Resveratrol: A comprehensive review of its multifaceted health benefits, mechanisms of action, and potential therapeutic applications in chronic disease. Pharmacological Research - Natural Products, 2024. 3: p. 100047.

[14]        Sung, M.M., et al., Resveratrol treatment of mice with pressure-overload-induced heart failure improves diastolic function and cardiac energy metabolism. Circ Heart Fail, 2015. 8(1): p. 128-37.

[15]        Guo, S., Y. Zhou, and X. Xie, Resveratrol inhibiting TGF/ERK signaling pathway can improve atherosclerosis: backgrounds, mechanisms and effects. Biomed Pharmacother, 2022. 155: p. 113775.
